# Line-Field Confocal Optical Coherence Tomography Increases the Diagnostic Accuracy and Confidence for Basal Cell Carcinoma in Equivocal Lesions: A Prospective Study

**DOI:** 10.3390/cancers14041082

**Published:** 2022-02-21

**Authors:** Charlotte Gust, Sandra Schuh, Julia Welzel, Fabia Daxenberger, Daniela Hartmann, Lars E. French, Cristel Ruini, Elke C. Sattler

**Affiliations:** 1Department of Dermatology and Allergy, University Hospital, LMU Munich, Frauenlobstr. 9-11, 80337 Munich, Germany; charlotte.gust@t-online.de (C.G.); fabiadaxenberger@gmail.com (F.D.); daniela.hartmann@med.uni-muenchen.de (D.H.); lars.french@med.uni-muenchen.de (L.E.F.); 2Department of Dermatology and Allergy, University Hospital, 86179 Augsburg, Germany; sandra.schuh@uk-augsburg.de (S.S.); julia.welzel@uk-augburg.de (J.W.); 3Dr. Phillip Frost Department of Dermatology and Cutaneous Surgery, Miller School of Medicine, University of Miami, Miami, FL 33125, USA; 4PhD School in Clinical and Experimental Medicine, University of Modena and Reggio Emilia, 41121 Modena, Italy

**Keywords:** basal cell carcinoma, dermoscopy, line-field confocal optical coherence tomography, bedside histology, skin imaging, non-invasive diagnostics in dermatology

## Abstract

**Simple Summary:**

Basal cell carcinoma is the most frequently occurring type of skin cancer. Its treatment can be either local or surgical depending on its subtype and extension, with early recognized and superficial cases being easier to treat. Some of them, however, display unspecific features, making diagnosis difficult. Non-invasive devices such as line-field confocal optical coherence tomography (LC-OCT) are able to recognize morphological features of different BCC subtypes with a good correlation to histopathology. We decided to study their application to clinically doubtful BCC cases.

**Abstract:**

Diagnosing clinically unclear basal cell carcinomas (BCCs) can be challenging. Line-field confocal optical coherence tomography (LC-OCT) is able to display morphological features of BCC subtypes with good histological correlation. The aim of this study was to investigate the accuracy of LC-OCT in diagnosing clinically unsure cases of BCC compared to dermoscopy alone and in distinguishing between superficial BCCs and other BCC subtypes. Moreover, we addressed pitfalls in false positive cases. We prospectively enrolled 182 lesions of 154 patients, referred to our department to confirm or to rule out the diagnosis of BCC. Dermoscopy and LC-OCT images were evaluated by two experts independently. Image quality, LC-OCT patterns and criteria, diagnosis, BCC subtype, and diagnostic confidence were assessed. Sensitivity and specificity of additional LC-OCT were compared to dermoscopy alone for identifying BCC in clinically unclear lesions. In addition, key LC-OCT features to distinguish between BCCs and non-BCCs and to differentiate superficial BCCs from other BCC subtypes were determined by linear regressions. Diagnostic confidence was rated as “high” in only 48% of the lesions with dermoscopy alone compared to 70% with LC-OCT. LC-OCT showed a high sensitivity (98%) and specificity (80%) compared to histology, and these were even higher (100% sensitivity and 97% specificity) in the subgroup of lesions with high diagnostic confidence. Interobserver agreement was nearly perfect (95%). The combination of dermoscopy and LC-OCT reached a sensitivity of 100% and specificity of 81.2% in all cases and increased to sensitivity of 100% and specificity of 94.9% in cases with a high diagnostic confidence. The performance of LC-OCT was influenced by the image quality but not by the anatomical location of the lesion. The most specific morphological LC-OCT criteria in BCCs compared to non-BCCs were: less defined dermoepidermal junction (DEJ), hyporeflective tumor lobules, and dark rim. The most relevant features of the subgroup of superficial BCCs (sBCCs) were: string of pearls pattern and absence of epidermal thinning. Our diagnostic confidence, sensitivity, and specificity in detecting BCCs in the context of clinically equivocal lesions significantly improved using LC-OCT in comparison to dermoscopy only. Operator training for image acquisition is fundamental to achieve the best results. Not only the differential diagnosis of BCC, but also BCC subtyping can be performed at bedside with LC-OCT.

## 1. Introduction

In recent decades, basal cell carcinoma (BCC) has progressed towards being the most frequently occurring type of skin cancer [1,2]. The diagnostic gold standard includes a biopsy or excision of the suspicious lesion and histopathological examination, resulting in a cost- and time-intensive process [3,4]. Treatment depends mainly on the histopathological subtype, with deep and nodular or infiltrating BCCs requiring a complete surgical excision, while superficial tumors can benefit from cryotherapy, lasers, or topical drugs such as imiquimod [2,3]. Therefore, the bedside diagnosis and subtyping of BCC are crucial for treatment planning. Clinical and dermoscopical examinations are commonly used in the daily clinical practice to diagnose BCCs, but non-invasive optical diagnostic methods, such as optical coherence tomography (OCT) and reflectance confocal microscopy (RCM), have shown a high potential in early detection of clinically unclear BCCs, with diagnostic sensitivity and specificity over 90% [5,6,7]. Additionally, the new line-field confocal OCT (LC-OCT), with higher penetration depth than RCM (500 µm compared to 250 µm) and higher resolution than OCT (1 µm compared to 7.5–10 µm) [8,9], has recently been used to detect BCCs non-invasively according to morphological criteria in preliminary studies [10,11,12]. However, there are no studies focused on clinically unclear lesions.

The aims of this study were: to verify in a real-life setting the diagnostic value of LC-OCT for clinically unclear BCC cases and to test for the ability to distinguish between superficial and other BCC subtypes, compared to dermoscopy and histopathology. Moreover, we reviewed morphological LC-OCT criteria for BCCs and provide an overview of the main diagnostic pitfalls. 

## 2. Materials and Methods

This prospective study was performed at the Departments of Dermatology of the University of Munich and the University of Augsburg in Germany between November 2019 and February 2021. We prospectively recruited 182 clinically unclear lesions of 154 patients that were referred to our imaging departments from the general outpatient and inpatient departments and from external specialists in order to confirm or to rule out BCC prior to biopsy or excision. Clinical, dermoscopic (FotoFinder GmbH, Germany and Dermogenius-Dermoscan GmbH, Regensburg, Germany), and LC-OCT images of the cases were acquired and analyzed by two blinded imaging experts (CR, CG). Discordant cases were reviewed by a third expert (ES, SS, or JW). After the non-invasive imaging, surgical excision followed by histopathological examination was performed in all lesions. 

For every case, the patient’s age, sex, and the anatomical site of the lesion were registered. Afterwards, dermoscopy was performed and a dermoscopic diagnosis was registered together with a confidence level ranging from 1 = high (>75% confidence) to 2 = intermediate (>50% confidence) to 3 = low (<50% confidence). Finally, the lesions were scanned with the LC-OCT device (DAMAE Medical, Paris), which uses a class 1 supercontinuum laser with a central wavelength of 800 nm to create various A-scans up to a depth of approximately 500 µm. Three imaging modalities are available: vertical or en-coupe, horizontal or en-face, and a 3D reconstruction either in a vertical or horizontal field of view. Details are described elsewhere [8,9].

The following features were registered: LC-OCT diagnosis, confidence, image quality, and whether LC-OCT could rule out a BCC diagnosis or not. LC-OCT image quality was scored from 1 (high: perfect image, no artefacts) to 2 (low: lower resolution, minor artefacts) to 3 (insufficient: major artefacts, unreadable). Confidence level was measured as described above for dermoscopy.

We calculated the frequency of the main descriptive morphological parameters used for the identification of BCCs in LC-OCT and their frequency as previously described in published studies [11] (Table 1). Key features useful for differentiating BCCs from non-BCCs and to differentiate between superficial BCC and other subtypes were identified by logistic regression. 

To assess the LC-OCT performance, diagnostic accuracy, sensitivity, and specificity for BCCs compared to dermoscopy and to the gold standard histology were calculated using McNemar’s tests. Calculation was performed for all lesions and additionally in the subgroup of cases with a confidence level rated as “high” by at least one of the observers. 

The study was approved by the ethical committee of the LMU Munich (Protocol Number 17-699).

## 3. Results

### 3.1. Population

A total of 182 lesions clinically suspicious for BCC in 154 patients were included in this study; 113 (62.1%) were confirmed histopathologically as BCCs and 69 (37.9%) as non-BCCs. The histopathological analysis entailed 35 (19.2%) completely excised lesions, 132 (72.5%) lesions acquired by punch biopsy, and 15 (8.2%) shave biopsies. Among the 113 BCCs, the nodular BCC subtype was seen most frequently in 52 (46%) lesions, followed by the superficial subtype in 35 (31%) and 4 (3.5%) infiltrative cases. Among the non-BCC cases, actinic keratosis (AK) was the most common diagnosis with a total of 24 lesions equaling a prevalence of 34.8%. Additional diagnoses included: squamous cell carcinoma (SCC), Bowen’s disease, nevus, melanoma, sebaceous hyperplasia, scar tissue/fibrous papule, dermatofibroma, lentigo solaris/seborrheic keratosis, eczema, clear cell acanthoma, molluscum contagiosum, trichoblastoma, angioma, atypical fibroxanthoma, and granuloma (Figure 1, Figure 2, Figure 3, Figure 4 and Figure 5) (Table 2).

### 3.2. Image Quality and Diagnostic Confidence

Average LC-OCT image quality was rated as high in 162 cases (89%) and as low in 20 cases (11%); no image was considered unreadable (image quality 3).

A high diagnostic confidence (1) by at least one observer was only scored in 48% of the dermoscopic images, compared to 70% of the LC-OCT cases. The interobserver agreement on the confidence level was very good. With LC-OCT, this was the case in 173 out of 182 (95%) of the observations (weighted kappa = 0.90 (95% CI, 8.84 to 0.97), *p*~0). For dermoscopy, the number of interobserver agreements was 177 out of 182 (97.3%) observations (weighted kappa = 0.96 (95% CI, 0.92 to 0.99), *p*~0).

Our average diagnostic confidence with LC-OCT was 1.7; if we consider the subgroup of lesions with high image quality (1), we reported a confidence level of 1.1 (limbs) and 1.2 (head, trunk); in the subgroup of lesions with low image quality (2), diagnostic confidence sank to 2.3 (trunk), 2.5 (head), and 2.8 (limbs). Diagnostic confidence was therefore influenced by image quality but not by anatomical site.

### 3.3. Diagnostic Performance

Diagnostic accuracy of dermoscopy and LC-OCT for ruling out BCCs in suspicious lesions was 88% and 91%, respectively. Concerning sensitivity and specificity compared to the gold standard histology, dermoscopy scored a sensitivity of 90% and a specificity of 86%, while LC-OCT showed a high sensitivity (98%) and a good but lower specificity (80%). Sensitivity and specificity slightly increased when image quality was high, to 93% and 99%, respectively. If considering the subgroup of LC-OCT images with a high diagnostic confidence, however (70% of the lesions), the LC-OCT performance increased significantly, with a sensitivity of 100% and a specificity of 97% compared to the gold standard histology (*p* = 1, McNemar’s test).

In daily clinical practice, the combination of dermoscopy and LC-OCT enables the clinician to confidently make an almost perfect diagnosis at the bedside. The combination of both reaches an accuracy of 92.9% in all cases and an accuracy of 98.2% in confident cases, providing a certain security for the clinician and guiding further steps in the treatment process. 

### 3.4. BCC Subtype

The diagnostic accuracy of LC-OCT for all BCC subtypes was 90%; we reported an overall sensitivity of 77% and a specificity of 96%. When only looking at cases with either a high LC-OCT quality (n = 101, 11% of the lesions) or lesions with a high LC-OCT confidence level (n = 93, 84% of the lesions), the performance of LC-OCT showed no significant difference to histology for all subtypes (*p* > 0.05, McNemar’s test). Diagnostic accuracy, sensitivity, and specificity for each subtype were: superficial BCC, 90%, 77%, 96%; nodular BCC, 88%, 96%, 82%; mixed BCC, 90%, 67%, 96%. This values slightly increased when selecting only high image quality (Table 3).

If we consider only pure BCC subtypes (86 lesions), the diagnostic accuracy increased to 92%. Sensitivity and specificity were for superficial BCCs 82% and 100%, for nodular BCCs 100% and 81%, and for fibrosing BCCs 75% and 100%, respectively. 

### 3.5. Diagnostic Criteria

The following nine parameters were entered in the first model of the logistic regression: vertical parameters: hyperkeratosis, thinning of the epidermis, poorly defined DEJ, hyporeflective ovoid structures, dark rim/clefting; horizontal parameters: polarization of nuclei in the epidermis, palisading, clefting, and tumor nests. Through a backward elimination approach, non-significant parameters were removed one by one so that the three most impactful parameters to discriminate between BCC and non-BCC were: poorly defined DEJ, dark rim/clefting, and hyporeflective ovoid structures. These findings were in line with previous work [10,11].

Concerning BCC subtypes, following parameters were inserted in the logistic regression: thinning of the epidermis, shoal of fish pattern of the lobules, and string of pearls pattern of the lobules. After logistic regression, thinning of the epidermis and string of pearls were the most impactful key criteria to influence the distinction between superficial BCC and other BCC subtypes (Figure 1, Figure 2 and Figure 3).

### 3.6. Other Diagnoses

With dermoscopy, 10 lesions were mistakenly classified as BCC: two AK, two sebaceous hyperplasias (Figure 5), one Bowen’s disease, one SCC, one nevus, one dermatofibroma, one molluscum contagiosum (Figure 1), and one pyogenic granuloma. Eleven BCCs were instead classified as: five SCC, two AK, one lentigo maligna/melanoma, one nevus, one scar, and one dermatofibroma. Using LC-OCT, we reported 14 false positive cases: eight AK, two seborrheic keratosis, one Bowen’s disease, one sebaceous hyperplasia, one molluscum contagiosum, and one pyogenic granuloma. Two cases of nodular BCCs were mistakenly diagnosed as a scar and an SCC. 

Keratinocyte skin cancer (KC), consisting of actinic keratoses (AKs), squamous cell carcinoma (SCC), and Bowen carcinomas (BCs), made up 38 of the 182 suspicious lesions. The performance of LC-OCT differed on KC compared to BCCs, showing a change in sensitivity, specificity, and confidence. Compared to dermoscopy, LC-OCT was more specific but less sensitive (N = 182, McNemar’s *p* value = 0.02).

## 4. Discussion

In this study, we aimed to analyze the diagnostic performance of LC-OCT for BCCs among clinically unclear lesions, which can be missed with dermoscopy since they display unspecific patterns. We determined our diagnostic confidence and our diagnostic accuracy, sensitivity, and specificity in comparison to dermoscopy and to the gold standard histology. Being in a real-life setting, we also focused on our capability of distinguishing superficial BCCs from other subtypes using LC-OCT, since this has a direct influence on the choice of the therapy. Moreover, we tried to identify morphological criteria specific for BCC compared to other non-BCC differential diagnoses, which could be helpful to avoid unnecessary biopsies in the case of a benign diagnosis and to direct the patient to the correct therapeutic path in the case of malignant lesions other than BCC. 

### 4.1. BCC Morphologic Criteria and Comparison with Histology

While clinical examination and dermoscopy are able to diagnose most clear-cut BCCs, equivocal lesions are a special target of non-invasive diagnostic techniques [5]. To date, a few preliminary studies have described morphological criteria for diagnosing BCC using the novel device LC-OCT, with good histopathological correlations [10,11]. Suppa et al. examined 89 BCCs and described tumor lobules, with (in superficial BCCs) or without connection (in nodular BCCs) to the epidermis and branched lobules (in infiltrative BCCs). In our previous study [11], we reported overlapping findings and defined, analogously to OCT terminology, the presence of a string of pearls pattern in superficial BCCs and of a shoal of fish pattern in infiltrative BCCs. Moreover, we reported an overall BCC subtype agreement between LC-OCT and histology of 90.4%, with a sensitivity of 82% and a specificity of 100% for superficial BCCs.

### 4.2. Diagnostic Accuracy, Sensitivity, and Specificity

Our diagnostic accuracy for BCCs was very high with both dermoscopy (88%) and LC-OCT (91%); compared to the gold standard histology, LC-OCT reached a high sensitivity (98%) but a slightly lower specificity (80%) in contrast to dermoscopy (86%). 

Nevertheless, LC-OCT’s specificity was higher than shown in recent studies on the use of other diagnostic techniques: OCT (range: 73–75%) or RCM (range: 38–59%). Such tools have been used in similar studies to better characterize unclear lesions. Ulrich et al., for example, analyzed the sensitivity and specificity of OCT for the assisted diagnosis of non-pigmented BCC in a similar way in a previous observational study [13]. The additional use of OCT resulted in a specificity of 75.3% (LC-OCT: 80%), a sensitivity of 95.7% (LC-OCT: 98%), and an accuracy of 87.4% (LC-OCT: 92.9%). 

The combination of dermoscopy and LC-OCT reached, in our study, an accuracy of 92.9% in all cases and of 98.2% in confident cases, providing a certain security for the clinician and guiding further steps in the treatment process. Using both tools in clinical practice enables the clinician to confidently make a very accurate diagnosis at the bedside. 

Dermoscopy alone scored a lower sensitivity (90%) than expected [14]. This was probably due to the setting of clinically equivocal lesions, further supported by the fact that for only 48% of the lesions examined, the clinicians had a high level of confidence in their diagnosis with dermoscopy. This finding supports the fact that the enrolled lesions were equivocal. Our confidence improved markedly (70%) using LC-OCT. 

Interestingly, if we consider only lesions with a high level of diagnostic confidence with LC-OCT (the majority), sensitivity and specificity values for the diagnosis of BCC reach those of the gold standard histology (100% and 97%, McNemar’s *p* value = 1). 

### 4.3. Diagnostic Confidence

The abovementioned findings point out that the clinician can significantly increase diagnostic confidence after dermoscopy based on LC-OCT images and avoid a potentially unnecessary biopsy if reassured by high confidence (which was the case in most lesions in our study). The potential practical consequences are the reduction in pain and discomfort for the patient and surgery related costs and complications.

Our diagnostic confidence was not influenced by the anatomical site of the examined lesions, but was negatively influenced by the image quality. In fact, diagnostic confidence was significantly lower (1.2 vs. 2.5) in the subgroup of high vs. low image quality. This emphasizes the value of correct and standardized image acquisition protocols and required training for imaging performing personnel.

### 4.4. BCC Subtyping

After correctly diagnosing a BCC out of clinically equivocal lesions, the next step requires its classification into the superficial or non-superficial subtypes; in fact, a superficial BCC can be easily treated with topical drugs or further local treatments, while other more invasive subtypes should be referred to (micrographically controlled) surgery. 

We identified the absence of the thinning of the epidermis and a string of pearls pattern with lobules connected to the DEJ as the most useful criteria for diagnosing superficial BCCs. The string of pearls pattern, present in 80% of our superficial BCCs, was described as hemispheric lobules (63%) or lobules connected to the epidermis (100%) in Suppa et al.’s findings. We demonstrated that we are able to use LC-OCT to distinguish between superficial and non-superficial BCC subtypes with high diagnostic accuracy. We reached a sensitivity and specificity for diagnosing superficial BCCs of 77% and 96%, respectively, increasing to 82% and 100% when considering only pure subtypes [11].

Mixed BCC subtypes are a diagnostic challenge, since their components vary in their amount and can be easily missed. However, when nodular or fibrosing components are present, the BCC becomes a surgery candidate and should not be (except for very superficial nodular components) treated with less invasive methods. Therefore, superficial BCCs could be treated non-invasively after LC-OCT without histopathological examination. 

For nodular BCC detection, LC-OCT showed the highest sensitivity with 100%, whereas specificity was highest for superficial (100%) and infiltrating (100%) BCCs. However, it needs to be mentioned that for the infiltrating BCC subtype, only a few lesions (n = 4) were collected in the real-life setting and the category classified as mixed subtype includes nodular superficial, nodular fibrosing, and other mixed subtypes. This may be seen as a limitation of this study.

Again, the performance of LC-OCT for all subtypes including mixed subtypes showed no significant difference to histology for all subtypes (*p* > 0.05, McNemar’s test) when only looking at cases with either a high LC-OCT quality (n = 101, 11% of the lesions) or lesions with a high LC-OCT confidence level (n = 93, 84% of the lesions). 

In a second step, we tried to identify with the help of logistic regressions a few key criteria able to distinguish between BCC and non-BCCs; these were: poorly defined DEJ, hyporeflective ovoid structures or nests/lobules (Figure 3), and dark rim/clefting (Figure 1). These criteria were consistent with previous findings [10,11].

### 4.5. Differential Diagnosis and Related Pitfalls

Although LC-OCT is a promising tool for the bedside differential diagnosis of BCC, there still remain pitfalls. Fourteen lesions (eight actinic keratosis, two lentigo solaris/seborrheic keratosis, one Bowen’s disease, one sebaceous hyperplasia, one molluscum contagiosum, one pyogenic granuloma) were misdiagnosed as BCCs. 

The actinic keratosis belonged to the bowenoid subtype, causing a roundish contour of the epidermis with atypical keratinocytes that can be confounded with a string of pearls pattern. In this case, the expert observer should pay attention to the bright hyperkeratosis and continuity of the DEJ, Analogously, tumor lobules can be easily caused by granulomas, sebaceous lobules [15], and even molluscum bodies [16]. In such cases, the roundish or polycyclic dermal lobules are very well defined and sharply contoured, and usually contain brighter granular structures (Figure 4: molluscum contagiosum, Figure 5: sebaceous hyperplasia) or fibrotic–calcific tissue (granuloma). Furthermore, the DEJ is usually preserved and is overlined by a normal epidermis, which can, however, be thinned by the dermal structures. Sebaceous hyperplasia is also usually connected to hair follicles.

One case of BCC was misdiagnosed as a scar, probably due to the hyperreflective connective tissue masking the presence of tumor lobules, while another one was classified as an SCC due to the surrounding field cancerization in an elderly patient with hyperkeratosis, keratinocyte atypia, elastosis, and collagen alterations. For this reason, particular attention is needed in such patients when performing a bedside mapping of multiple suspicious lesions.

## 5. Conclusions

To sum up, we were able to reach a very good diagnostic confidence and performance in distinguishing BCCs from other BCC-suspicious lesions compared to dermoscopy and the gold standard histology. The study had limitations such as the small number of non-BCC lesions and the presence of only histologically confirmed lesions. Nevertheless, we believe LC-OCT is able to support the clinician in the process of diagnosing and subtyping BCC, as a key role to optimize the diagnostic approach and the treatment. It is particularly useful to screen BCCs in the context of clinically equivocal lesions, which can be difficult to diagnose with dermoscopy only. Moreover, it is possible to screen for superficial BCCs to be treated with less invasive methods than surgery. Larger studies on specific lesion subgroups such as facial papules are needed to gain more experience in this interesting field of dermatology.

## Figures and Tables

**Figure 1 cancers-14-01082-f001:**
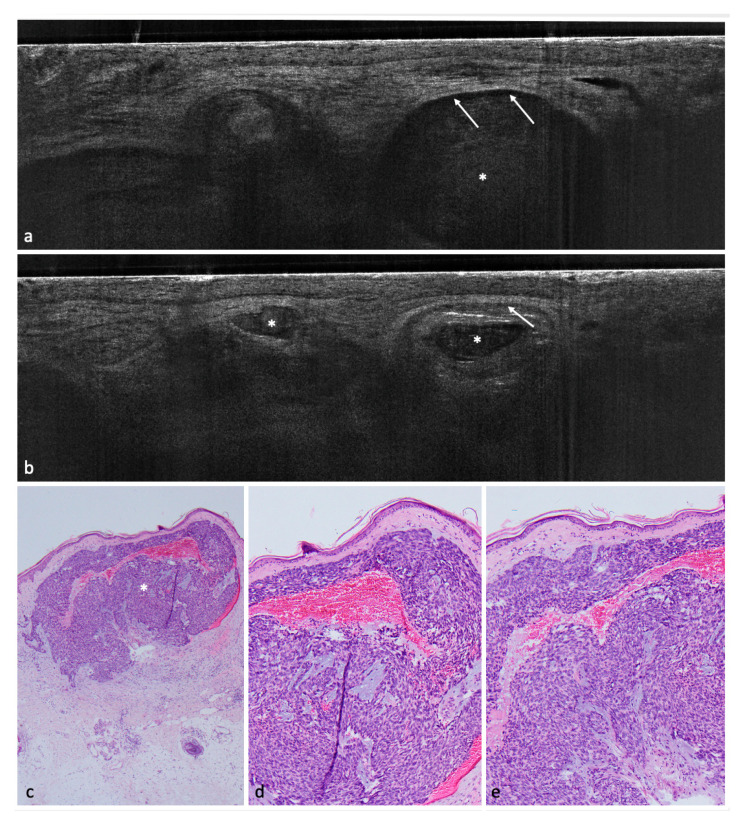
Nodular BCC on the lower leg of a 68-year-old female patient. (**a**,**b**) LC-OCT images. The nodular BCC presents itself with a fine granular texture corresponding to basaloid cells, peritumoral clefting (white arrows) and homogeneous areas with possibly liquefactive necrosis with remaining cell debris (white asterisks). (**c**) (40×), (**d**) (100×), and (**e**) (100×): corresponding histological HE-stained sections with peripheral palisading, clefting, and a central necrosis.

**Figure 2 cancers-14-01082-f002:**
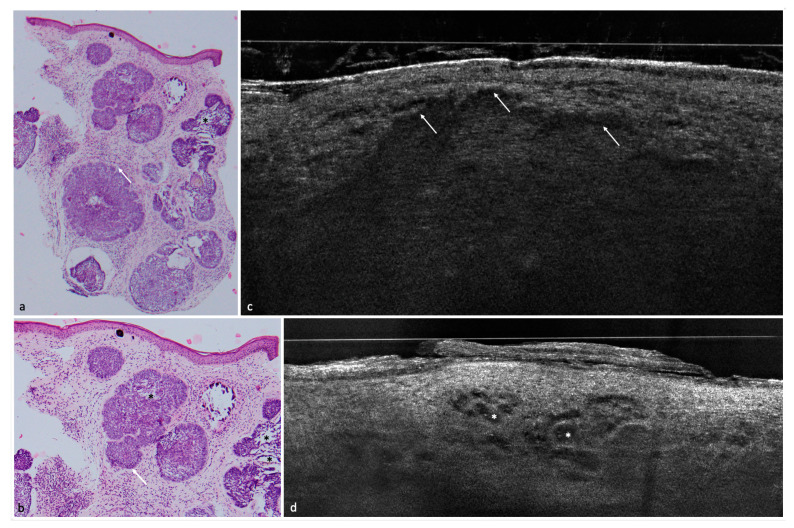
Nodular BCC with cystic parts in the nasolabial fold of a 76-year-old patient. (**a**) (200×) and (**b**) (100×): histological HE-stained sections. Peritumoral clefting (white arrows) and cystic structures (asterisk). (**c**) LC-OCT image of the same lesion. Arrows again indicate clefting, the BCC has a fine granular texture. (**d**) dark hyporeflective area (asterisk) in a cystic BCC corresponding to glomerular vessels.

**Figure 3 cancers-14-01082-f003:**
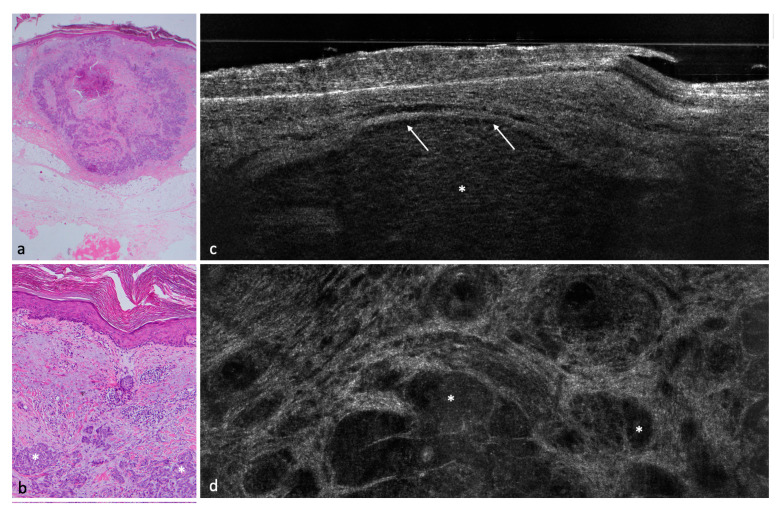
Nodular BCC on the right temple of a 64-year-old male patient. (**a**) (20×) and (**b**) (100×): histological HE-stained sections of the tumor. (**c**) (vertical) and (**d**) (horizontal): LC-OCT images of the corresponding lesion. Again, peripheral clefting can be seen (white arrow) as well as basaloid nests (asterisks).

**Figure 4 cancers-14-01082-f004:**
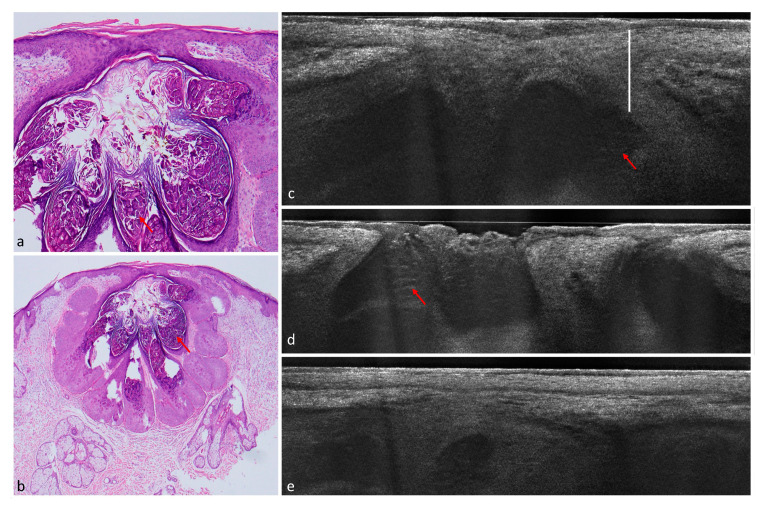
Molluscum contagiosum on the forehead of a 58-year-old male patient. (**a**) (100×) and (**b**) (40×): histological hematoxylin and eosin (HE)-stained sections. In histology, a cup-shaped multilobular lesion can be identified. Red arrows indicate intracytoplasmatic inclusion bodies (Henderson–Paterson bodies) which contain virus particles. (**c**–**e**) LC-OCT image of the molluscum contagiosum. It shows acanthosis (white bar) and hyporeflective areas with hazy structures, which probably correspond to the Henderson–Paterson bodies (red arrow).

**Figure 5 cancers-14-01082-f005:**
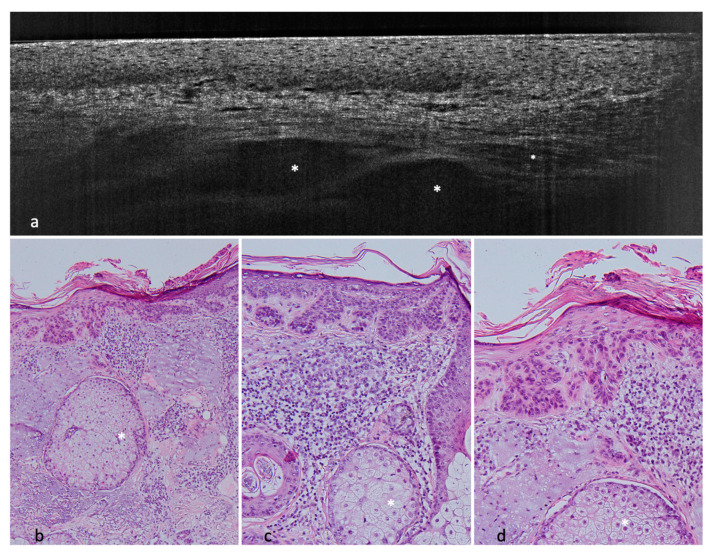
Sebaceous hyperplasia on the nose of a 70-year-old male patient. (**a**) LC-OCT image. The enlarged sebaceous glands are visible as hyporeflective ovoid structures (white asterisk). When looking carefully, roundish sebocytes might be spotted within the glands. (**b**) (100×) and (**c**) (200×) and (**d**) (200×): histological HE-stained sections. Enlarged, regularly structured sebaceous glands (white asterisk).

**Table 1 cancers-14-01082-t001:** LC-OCT parameters.

(**a**) Epidermal LC-OCT parameters vertical												
**EPIDERMAL LC-OCT PARAMETERS VERTICAL**		**Parameters used in logistic regression for:**		**Not-BCC** **(n = 69)**	**All BCC** **(n = 113)**	**Superficial BCC** **(n = 35)**	**Nodular BCC** **(n = 52)**	**Infiltrative BCC** **(n = 4)**			**Mixed BCC** **(n = 21)**	
		**BCC vs. non BCC**	**Superficial BCC vs. other pure BCC subtypes (nodular, fibrosing)**									
Hyperkeratoses		x		40 (58.0)	38 (33.6)	13 (37.1)	20 (38.5)	1 (25.0)			4 (19.0)	
Thinning of the epidermis		x	x	5 (7.2)	48 (42.5)	7 (20.0)	34 (65.4)	1 (25.0)			6 (28.6)	
Bowenoid morphology				14 (20.3)	5 (4.4)	3 (8.6)	2 (3.8)	0 (0)			0 (0)	
Acanthosis, verrucous surface				11 (15.9)	4 (3.5)	1 (2.9)	2 (3.8)	0 (0)			1 (4.8)	
Scales				39 (56.5)	34 (30.1)	12 (34.3)	14 (26.9)	2 (50.0)			6 (28.6)	
Keratin plugs/horn cysts				17 (24.6)	0 (0)	0 (0)	0 (0)	0 (0)			0 (0)	
Ulceration				15 (21.7)	35 (31)	9 (25.7)	17 (32.7)	2 (50.0)			7 (33.3)	
Pagetoid cells				3 (4.3)	1 (0.9)	0 (0)	1 (1.9)	0 (0)			0 (0)	
Ovoid concentric structures (sebaceous gland)				19 (27.5)	7 (6.2)	0 (0)	4 (7.7)	1 (25.0)			2 (9.5)	
Atypical junctional nests				2 (2.9)	1 (0.9)	0 (0)	1 (1.9)	0 (0)			0 (0)	
Tumour strands				6 (8.7)	2 (1.8)	0 (0)	2 (3.8)	0 (0)			0 (0)	
Wave pattern/junctional nests				5 (7.2)	1 (0.9)	0 (0)	0 (0)	1 (25.0)			0 (0)	
Dermal/deep nests				4 (5.8)	0 (0)	0 (0)	0 (0)	0 (0)			0 (0)	
Atypical honeycombed pattern				49 (71.0)	74 (65.5)	19 (54.3)	39 (75.0)	3 (75.0)			13 (61.9)	
(**b**) DEJ—dermal parameters vertical												
**DEJ-DERMAL PARAMETERS VERTICAL**		**Flattened rete ridges**	**Elongated rete ridges**	**Well defined DEJ**	**Less defined (disrupted) DEJ**	**Hyporefrective ovoid structures/nests/lobules**	**Dark rim/** **clefting**	**Prominent vessels**	**Shoal of fish pattern of the lobules**	**String of pearls patterns of the lobules (connected to the DEJ)**	**White stromal reaction (vertical)**	**Black areas/hyporeflective cysts inside the nests**
Parameters used in logistic regression for:	BCC vs. non BCC				X	X	X					
	Superficial BCC vs. other pure BCC subtypes (nodular, fibrosing)								X	X		
Not-BCC	(n = 69)	6 (8.7)	2 (2.9)	41 (59.4)	21 (30.4)	10 (14.5)	9 (13.0)	28 (40.6)	3 (4.3)	8 (11.6)	21 (30.4)	3 (4.3)
All BCC	(n = 113)	1 (0.9)	0 (0)	26 (23.0)	107 (94.7)	107 (94.7)	88 (77.9)	54 (47.8)	26 (23.0)	56 (49.6)	64 (56.6)	19 (16.8)
Superficial BCC	(n = 35)	0 (0)	0 (0)	10 (28.6)	34 (97.1)	32 (91.4)	29 (82.9)	14 (40.0)	13 (37.1)	28 (80.0)	16 (45.7)	1 (2.9)
Nodular BCC	(n = 52)	0 (0)	0 (0)	11 (21.2)	50 (96.2)	51 (98.1)	38 (73.1)	25 (48.1)	4 (7.7)	13 (25.0)	31 (59.6)	12 (23.1)
Infiltrative BCC	(n = 4)	1 (25.0)	0 (0)	1 (25.0)	3 (75.0)	2 (50.0)	1 (25.0)	4 (100)	4 (100)	0 (0)	3 (75)	0 (0)
Mixed BCC	(n = 21)	0 (0)	0 (0)	3 (14.3)	20 (95.2)	21 (100)	19 (90.5)	11 (52.4)	5 (23.8)	15 (71.4)	14 (66.7)	6 (28.6)
(**c**) LC-OCT parameters horizontal												
**PARAMETERS HORIZONTAL**		**Pagetoid cells in the epidermis-DEJ (horizontal)**	**Bright stromal reaction (horizontal)**		**Polarization of nuclei in the epidermis (horizontal)**	**Palisading (horizontal)**	**Clefting (horizontal)**	**Tumor nests/cords/silhouettes (horizontal)**			**Prominent vessels (horizontal)**	**Bright collagen alterations (elastosis)(horizontal)**
Parameters used in logistic regression for:	BCC vs. non BCC				X	X	X	X				
	Superficial BCC vs. other pure BCC subtypes (nodular, fibrosing)											
Not-BCC	(n = 69)	4 (5.8)	17 (24.6)		2 (2.9)	1 (1.4)	3 (4.3)	10 (14.5)			9 (13.0)	30 (43.5)
All BCC	(n = 113)	28 (24.8)	70 (61.9)		58 (51.3)	40 (35.4)	38 (33.6)	78 (69.0)			42 (37.2)	79 (69.9)
Superficial BCC	(n = 35)	13 (37.1)	18 (51.4)		17 (48.6)	10 (28.6)	10 (28.6)	25 (71.4)			16 (45.7)	21 (60.0)
Nodular BCC	(n = 52)	11 (21.2)	34 (65.4)		28 (53.8)	23 (44.2)	23 (44.2)	37 (71.2)			16 (30.8)	43 (82.7)
Infiltrative BCC	(n = 4)	0 (0)	3 (75.0)		2 (50.0)	1 (25.0)	1 (25.0)	3 (75.0)			2 (50.0)	3 (75.0)
Mixed BCC	(n = 21)	4 (19.0)	15 (71.4)		10 (47.6)	6 (28.6)	4 (19.0)	13 (61.9)			8 (38.1)	11 (52.4)

**Table 2 cancers-14-01082-t002:** Epidemiological data.

Characteristic	Patient-Based	Lesion-Based
Mean age (+/−SD)	70.8 (12.3)	71.7 (12.0)
Sex, n(%)		
Male	102 (66.2)	
Female	52 (33.8)	
Localisation, n (%)		
Ear	1 (0.6)	1 (0.5)
Face	76 (49.4)	85 (46.7)
Head	4 (2.6)	6 (3.3)
Neck	3 (1.9)	4 (2.2)
Scalp	3 (1.9)	4 (2.2)
Trunk	49 (31.8)	62 (34.1)
Lower limb	9 (5.8)	10 (5.5)
Upper limb	9 (5.8)	10 (5.5)
Number of lesions n (%)		
1	132 (85.7)	
2	17 (11.0)	
3	4 (2.6)	
4	1 (0.6)	
Histological diagnosis, n (%)		
*BCC*	91 (59.1)	113 (62.1)
*non-BCC*	63 (40.9)	69 (37.9)
AK	21 (33.3)	24 (34.8)
SCC	8 (12.7)	9 (13.0)
Sebaceous hyperplasia	7 (11.1)	7 (10.1)
Bowen	4 (6.3)	5 (7.2)
Other non-BCC *	23 (36.5)	24 (34.8)
BCC subtypes, n (%)		
Superficial BCC	25 (27.5)	35 (31.0)
Nodular BCC	44 (48.4)	52 (46.0)
Fibrosing BCC	4 (4.4)	4 (3.5)
Nodular superficial BCC	12 (13.2)	15 (13.3)
Nodular Fibrosing BCC	2 (2.2)	3 (2.7)
Mixed	3 (3.3)	3 (2.7)

* Other non-BCC subtypes include lentigo maligna/melanoma, nevus, scar, dermatofibroma, lentigo solaris/seborrheic keratosis, eczema, clear cell acanthoma, molluscum contagiosum, fibrous papule, trichoblastoma, angioma, atypical fibroxanthoma, aspecific crust, and granuloma.

**Table 3 cancers-14-01082-t003:** LC-OCT performance.

(**a**) All lesions
All lesions	**LC-OCT**
BCC	Non-BCC	Total
Histology	BCC	111	2	113
Non-BCC	14	55	69
Total	125	57	182

	**Dermoscopy**
BCC	Non-BCC	Total
Histology	BCC	102	11	113
Non-BCC	10	59	69
Total	112	70	182
(**b**) High confidence lesions
High confidence level for at least one of the 2 observers	**LC-OCT**
BCC	Non-BCC	Total
Histology	BCC	93	0	93
Non-BCC	1	33	34
Total	94	33	127

	**Dermoscopy**
BCC	Non-BCC	Total
Histology	BCC	61	2	63
Non-BCC	0	24	24
Total	61	26	87
(**c**) Performance for BCC
**All BCC lesions**
N (%)	111 (100)
Global accuracy	84%
	Superficial BCC(N = 35)	Nodular BCC(N = 51)	Fibrosing BCC(N = 4)	Mixed(N = 21)
Accuracy	90%	88%	99%	90%
Sensitivity	77%	96%	75%	67%
Specificity	96%	82%	100%	96%
*p* value(Mac Nemar‘s test)	0.23	0.03	1	0.55
**Good image quality**
N (%)	101 (91)
Global accuracy	86%
	Superficial BCC(N = 31)	Nodular BCC(N = 49)	Fibrosing BCC(N = 2)	Mixed(N = 19)
Accuracy	91%	90%	100%	91%
Sensitivity	81%	96%	100%	68%
Specificity	96%	85%	100%	96%
P value(Mac Nemar‘s test)	0.51	0.11	-	0.50
	**High confidence level for at least one of the 2 observers**
N (%)	93 (84)
Global accuracy	84
	Superficial BCC(N = 31)	Nodular BCC(N = 43)	Fibrosing BCC(N = 1)	Mixed(N = 18)
Accuracy	89%	89%	100&	89%
Sensitivity	77%	95%	100%	67%
Specificity	95%	84%	100%	95%
*p* value(Mac Nemar‘s test)	0.34	0.11	-	0.75

## Data Availability

Fully anonymized data are available on motivated request.

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
