# Peer review of "Line-Field Confocal Optical Coherence Tomography Increases the Diagnostic Accuracy and Confidence for Basal Cell Carcinoma in Equivocal Lesions: A Prospective Study"

_cancers, 2022, doi:10.3390/cancers14041082_

Round 1

Reviewer 1 Report

Report on paper entitled “Line-field confocal optical coherence tomography increases the diagnostic accuracy and confidence for basal cell carcinoma in equivocal lesions: a prospective study”.

This paper reports on the performance of line-field confocal optical coherence tomography (LC-OCT) in recognizing morphologic features of different subtypes of difficult-to-diagnose Basal Cell Carcinoma
(BCCs). The results are compared with histopathology.

In general, the results are quite good for the diagnosis of BCC and the subtyping of superficial forms:
sensitivity of 98% and specificity of 80% for the diagnosis of BCC on the basis of 182 really equivocal lesions after clinical and dermoscopic examination. For the 70% of lesions with a high level of confidence in LC-OCT, the performance is almost perfect, with 100% sensitivity and 97% specificity.
Concerning subtyping, the specificity is very good on the superficial subtype (specificity of 96%). The overall discussion is very positive and shows superior performance to OCT (Vivosight).
The document is well written. The results are clearly presented. The images are described in detail.
The results are convincing.
I think this paper deserves to be published, with some modifications listed below:
***
- References 7 and 12 are identical.
- Line 69 (introduction): the resolution of the LC-OCT images is not 5µm.
- line 424-427 (conclusion): interest of this text? No relation with the paper.
- line 398: about molluscum contagiosum, I suggest to cite this reference: doi: 10.1111/jdv.17594.
- Figure 2: it is difficult to distinguish the sebocytes. The glands are quite deep and appear very
dilated.
- Figure 3: The LC-OCT image is a bit dark and not very resolved. Could you provide a better image?
- Figure 4: image d, asterisk, it looks strongly like glomerular vessels and not necrosis cavities.
- Figure 5 c: the arrow seems to be positioned on a horizontal vessel above the lobule and not on the
clefting of the lobule.
- Figure 6: This illustration of sebaceous hyperplasia is very unusual. I do not agree with the
interpretation because the sebaceous glands on the histology are very deep and therefore probably
not visible with LC-OCT and do not correspond to the asterisked areas on the LC-OCT image.
Moreover, I see no reason why some glands are hypo-reflective and others hyperreflective... In short,
I think it is not clear enough to deserve an illustration.

Author Response

Point by Point Response to the Reviewers

Dear Editors, dear Reviewers,

Thank you for your numerous and detailed comments. We took your feedback into consideration and worked on the manuscript in order to improve it and make it more suitable to publication. We believe our paper has now been significantly improved.

Reviewer Nr. 1

We are very thankful for your significant suggestions.

  • References 7 and 12 are identical.

Thank you for making us aware of our mistake. We removed the replication and adjusted the references accordingly.

  • Line 69 (introduction): the resolution of the LC-OCT images is not 5µm.

Thank you for pointing our inaccurate information on LC-OCT resolution out. Following reference 7, we adjusted the LC-OCT resolution to 1µm.

  • line 424-427 (conclusion): interest of this text? No relation with the paper.

Thank you for your counsel. We wanted to give an insight in potential new areas to use LC-OCT in the future, however we agree, it does not fit with the rest of our text and therefore we removed this part of our conclusion.

  • line 398: about molluscum contagiosum, I suggest to cite this reference: doi: 10.1111/jdv.17594.

Thank you for your advice. We added the suggested reference in order to provide further insight on LC-OCT usage in molluscum contagiosum.

  • Figure 2: it is difficult to distinguish the sebocytes. The glands are quite deep and appear very

Thank you. We adjusted the legend and image description correspondingly.

  • Figure 3: The LC-OCT image is a bit dark and not very resolved. Could you provide a better image?

Thank you for your counsel. We tried to improve the image quality in order to provide a better image with higher resolution and contrast.

  • Figure 4: image d, asterisk, it looks strongly like glomerular vessels and not necrosis cavities.

Thank you very much for pointing this out. We corrected the legend and added glomerular vessels to the description.

  • Figure 5 c: the arrow seems to be positioned on a horizontal vessel above the lobule and not on the clefting of the lobule.

Thank you. We totally agree, we put the arrow back into the originally meant spot on the clefting.

  • Figure 6: This illustration of sebaceous hyperplasia is very unusual. I do not agree with the interpretation because the sebaceous glands on the histology are very deep and therefore probably not visible with LC-OCT and do not correspond to the asterisked areas on the LC-OCT image. Moreover, I see no reason why some glands are hypo-reflective and others hyperreflective... In short, I think it is not clear enough to deserve an illustration.

Thank you. We reevaluated the figure and understand where you are coming from. Since it does not provide a typical representation of sebaceous glands in LC-OCT we went along and removed the figure from our work.

Reviewer 2 Report

The authors Gust et al. presented to the attention of the editors and reviewers quite a detailed paper on the topic of line-field confocal optical coherence tomography. This novel and not yet very much explored device was used in this study to investigate the diagnosis of basal cell carcinoma compared to dermoscopy. In contrast to other publications in which a comparison with histology and LC-OCT images was conducted, in order to describe imaging patterns, this study sets a focus on the subtype of clinically equivocal lesions, which are the ones that usually make clinicians struggle. The authors concluded that the use of LC-OCT provides a diagnostic advantage in difficult lesions compared to dermoscopy alone, especially when they evaluate good LC-OCT images and they recognize specific imaging patterns with high confidence.

I found the paper very interesting and with good

Elements of novelty. The methods

and the statistical analysis are correct and the questions to be answered are pragmatically

oriented. I consider this article worth of publication after it has been a little shortened and a few corrections are made.

- the simple summary and the abstract use sometimes repeated words. It would be stylistically nicer to differentiate them a little more.

- in the methods, you state that the most useful imaging mode was recorded, but there is no further mention of this aspect in the results. Either you remove it from the methods, or you further develop it in the results and discussion.

- The figures are really impressing, but I think their order should be optimized, in a more logical way. I would start with basal cell carcinomas and put the other diagnoses at the end.

-Table 1 and 3 need to be formatted in order to better fit the page; try the vertical orientation.

- the discussion should be shortened and the receptions avoided. I appreciate that you have a lot of endpoints and that you mentioned them in the first paragraph of the discussion in order to keep the reader focused; however, you could for example add subtitles and develop the discussion in the same order that you summarized in the first paragraph.

-The last paragraph of the conclusions is not correct and must be deleted.

- This original article has only 16 references; you should add some more references concerning similar Studies with other diagnostic techniques, such as https://doi.org/10.1159/000493727, 10.1016/j.jaad.2014.04.067. Epub 2014 Jun 11.,

Author Response

Point by Point Response to the Reviewers

Dear Editors, dear Reviewers,

Thank you for your numerous and detailed comments. We took your feedback into consideration and worked on the manuscript in order to improve it and make it more suitable to publication. We believe our paper has now been significantly improved.

Reviewer Nr. 2

Thank you for tremendously improving the quality of our manuscript.

1) the simple summary and the abstract use sometimes repeated words. It would be stylistically nicer to differentiate them a little more.

Thank you for your helpful suggestion. We tried to change our wording in order to make it more interesting and appealing to read.

2) in the methods, you state that the most useful imaging mode was recorded, but there is no further mention of this aspect in the results. Either you remove it from the methods, or you further develop it in the results and discussion.

Thank you for this important comment. We removed it from the methods since we have not developed it further since we saw that both imaging modes are useful for the global evaluation.

3) The figures are really impressing, but I think their order should be optimized, in a more logical way. I would start with basal cell carcinomas and put the other diagnoses at the end.

Thank you for your counsel. We followed your advice and rearranged the order of our figures, starting with BCC followed by the other diagnoses. 

4) Table 1 and 3 need to be formatted in order to better fit the page; try the vertical orientation.

Thank you. We went along and reformatted the tables 1 and 3 into a vertical orientation in an effort to better fit the page.

5) the discussion should be shortened and the receptions avoided. I appreciate that you have a lot of endpoints and that you mentioned them in the first paragraph of the discussion in order to keep the reader focused; however, you could for example add subtitles and develop the discussion in the same order that you summarized in the first paragraph.

Thank you for your comment, we appreciate your observations. We added subtitles and tried to arrange the discussion to better suit the initial order.

6) The last paragraph of the conclusions is not correct and must be deleted.

Thank you for making us aware of our misfitting paragraph. Since it does not entirely fit the message of this work, we removed the section.

7) This original article has only 16 references; you should add some more references concerning similar Studies with other diagnostic techniques, such as https://doi.org/10.1159/000493727, 10.1016/j.jaad.2014.04.067. Epub 2014 Jun 11.,

Thank you. We added the suggested references to provide further insight into similar studies and the usage of non invasive imaging modalities.

Round 2

Reviewer 1 Report

The revised paper can be published